# Occurrence of sensitive topics during ward round: an ancillary analysis of the BEDSIDE-OUTSIDE trial

Sebastian Gross,[1,2] Christoph Becker [1,3,4] , Katharina Beck,[1] Valentina Memma,[1] Jens Gaab,[2] Philipp Schütz,[4,5] Jörg D Leuppi,[4,6] Rainer Schaefert,[1,4,7] Wolf Langewitz [1,4] , Marten Trendelenburg,[4,8] Tobias Breidthardt,[4,8] Jens Eckstein,[4,8] Michael Osthoff,[4,8] Stefano Bassetti,[4,8] Sabina Hunziker [1,4,8]

For numbered affiliations see end of article.

**Correspondence to**
Professor Sabina Hunziker;
sabina.hunziker@usb.ch

## ABSTRACT

**Objective** Discussing sensitive topics (eg, medical uncertainty, social issues, non-adherence) during ward rounds is challenging and may negatively impact patient satisfaction with the healthcare they are receiving. In the previous multicentre randomised BEDSIDE-OUTSIDE trial focusing on communication during ward rounds, we investigated the interplay between sensitive topics and low reported satisfaction with care.

**Design** Pre-planned secondary analysis of a randomised controlled trial. For this analysis data of the original trial was pooled across intervention groups.

**Setting** Three Swiss teaching hospitals.

**Participants** Adult patients hospitalised for medical care.

**Interventions** We analysed predefined sensitive health topics and specific elements of communication from audiotapes recorded during ward rounds, for both patients dealing with and without sensitive topics.

**Primary and secondary outcome measures** The primary endpoint was overall patient satisfaction with care; measured on a Visual Analogue Scale from 0 to 100. Secondary endpoints included duration of ward rounds and further satisfaction outcomes.

**Results** Of the 919 included patients, 474 had at least one sensitive topic including medical uncertainty (n=251), psychiatric comorbidities (n=161), tumour diagnosis (n=137) and social issues (n=125). Compared with patients without sensitive topics, patients with sensitive topics reported lower satisfaction with care (mean (SD), 87.7 (±14.6) vs 90.2 (±12.1), adjusted difference −2.5 (95% CI −4.28 to −0.72), p=0.006. Among patients with sensitive topics, risk factors for low satisfaction included several parameters concerning patient–physician interaction such as disagreements during ward rounds (mean (SD), 14/212 (6.6%) vs 41/254 (16.1%), adjusted OR 2.78 (95% CI 1.47 to 5.27), p=0.002).

**Conclusions** A large proportion of medical inpatients must deal with sensitive health topics. This is associated with lower satisfaction with care, particularly if the patient perceives the interaction with doctors during ward rounds as unsatisfactory. Educating physicians on specific communication techniques may help improve care for these patients.

**Trial registration number** NCT03210987.

## STRENGTHS AND LIMITATIONS OF THIS STUDY

⇒ We investigated a large sample of a broad general medicine population from multiple centres.
⇒ Only Swiss teaching hospitals were included, which might cause selection bias and limit the generalisability of the findings.
⇒ Ward rounds of clinical practice procedures in the three participating hospitals were not standardised to ensure external validity—which may therefore reduce the internal validity.
⇒ The questionnaire used to assess patients' perceptions has not yet been validated.
⇒ As there is no universal definition for sensitive topics, different predefined sensitive topics have been pooled into a combined variable, which is a limitation of this study.

## INTRODUCTION

Life threatening disease, conflict with patients, patient non-adherence, psychiatric comorbidities, substance abuse, medical uncertainty among others are often considered to be sensitive topics. However, most research in this regard was mainly qualitative.[1] There is important work about the experiences of patients regarding bedside shift suggesting that asking patient consent, discussing potential critical issues and the degree of involvement preferred at hospital admission is strongly recommended.[2] Also, one quantitative analysis of mixed paediatric and adult patients including 622 ward round discussions found that social issues were addressed in 52%.[3] Thus further research is needed to understand frequency and implications of discussing sensitive topics during ward rounds in clinical practice. Sensitive topics may be important to address as they could affect the patient's well-being and include relevant healthcare information. On the other hand, the sharing of personal and sensitive information during clinical ward

rounds in a non-private setting could potentially offend patients who interpret this as a breach of doctor–patient confidentiality.[4] Physicians may avoid addressing sensitive topics, for example, by terminating problematical conversations, withholding frank responses, downplaying the patient's expressed emotions or inadequately acknowledging the sentiment underlying the patient's statements.[5 6]

A previous multicentre randomised trial including 919 internal medicine patients (BEDSIDE-OUTSIDE trial) aimed to compare the impact of patient case presentation during ward rounds on patients' medical knowledge. Patient case presentations were either conducted at the bedside or outside the patient room.[6] Compared with outside the room case presentation, bedside case presentation was shorter and resulted in similar patient knowledge, but importantly, sensitive topics were more often avoided and patient confusion was higher. Moreover, an ancillary analysis of the BEDSIDE-OUTSIDE trial focusing on staff satisfaction showed that physicians prefer to discuss sensitive topics outside the room than at the patient's bedside.[7] Despite concerns about addressing sensitive topics with patients, evidence in outpatient settings indicates that discussing these topics may be associated with high satisfaction, positive perceptions of healthcare, reduced worry and increased patient participation in treatment decisions in the outpatient setting.[8 9]

The current study aimed to quantitatively and qualitatively analyse audio tapes from internal medicine ward rounds of patients included in the BEDSIDE-OUTSIDE trial and to determine if discussing sensitive topics in clinical practice is associated with patient satisfaction.

## MATERIAL AND METHODS
### Study setting
The current study is a preplanned secondary analysis of the BEDSIDE-OUTSIDE trial[6]—a pragmatic, investigator-initiated, open-label, multicentre randomised trial conducted in the general medical divisions of three Swiss teaching hospitals (University Hospital Basel, Kantonsspital Aarau and Kantonsspital Baselland) between July 2017 and October 2019. This report adheres to the Consolidated Standards of Reporting Trials guidelines.[10]

This analysis studied potential risk factors associated with sensitive topics, the association of sensitive topics with different outcomes and potential risk factors for low or high levels of patient satisfaction with care. We also provided a qualitative overview of sensitive topic discussions during ward rounds. Because, to our knowledge, there is no well accepted definition of sensitive topics, we defined 'sensitive topics' based on the clinical experience of the physician–researcher team and by reviewing previous literature. Sensitive topics were coded prospectively as a situation where at least one of the following topics was discussed with the patient during the ward round: medical uncertainty, psychiatric comorbidities, tumour diagnosis, social issues, non-adherence, previous

conflicts between patient and treating team and treatment failure.

### Original study population
Newly admitted adult inpatients on medical wards expecting their first weekly ward round consultation were approached by a member of the study team regarding inclusion. Only one patient per room was eligible and we excluded individuals with cognitive or hearing impairment, those unable to understand the local language and patients who had previously been included in the study. All provided written informed consent.

### Study design, randomisation and intervention of the original study
Patients were randomised to either the 'bedside group' or the 'outside the room group' in a 1:1 ratio. In line with current practice, ward rounds for both groups followed the standard practices of each participating hospital.

Details of the study intervention and a detailed description of the ward round procedure have been reported earlier.[6] In brief, for the purpose of standardisation and in line with current practice in Switzerland, the ward round followed the routine medical ward round procedures in both groups, with defined roles of physicians and nurses per usual practice in each participating hospital. In the bedside presentation group, case presentations and discussions occurred only at the bedside in front of the patient, including clinical examination as appropriate, with no discussions beforehand. In the outside the room group, case presentation and discussions were primarily held in the hallway outside the room without the patient present. Afterwards, the team entered the room and gave the patient a short summary of the medical situation, completed the gathering of medical information, examined the patient as needed and discussed the next steps. Patients, study coordinators, and treating clinicians were not blinded to the allocation. However, study investigators involved in a patient's outcome assessment were blinded to trial allocation.

### Data collection
Data collection was conducted at different points in time. Baseline patient data was collected before the ward round. During the ward rounds, which were conducted between 09:00 and 11:00, an observer from the research team was present to document timing (ie, the duration of the ward round allocated per patient). All visits were recorded with an Apple iPad with the device-internal 'Voice Memo' software and rated afterwards by the research team. This approach allowed for the coding of various predefined sensitive topics and elements of communication. After the ward rounds, a second blinded member of the research team interviewed participating patients using a standardised questionnaire. All case report forms and items have been described in the original trial.[6]

## Patient and public involvement

A total of 25 patients hospitalised on the medical wards of the University Hospital of Basel, as well as 15 healthcare providers (physicians and nurses) with experience in daily medical practice and specially in medical ward rounds were involved in the design of the study and intervention of the main paper.[6] To design the trial, patients and healthcare providers helped us in prioritisation and selection of outcomes. Patients were asked for priority focus of this study. To design the intervention for the main trial, input was sought from patients and healthcare providers.

## Outcome measures

### Baseline factors and predictors

We assessed patient baseline characteristics including age, sex, number of children, family status, citizenship, level of education, occupation, main diagnosis and comorbidities. Predictors for the occurrence of sensitive topics included patients' quality of life assessed with the validated European Quality of Life 5 Dimensions 3 Level Version (EQ-5D-3L) questionnaire,[11] patient's Decisional Control Preference,[12 13] application of specific patient-centred communication techniques (WEMS: waiting, echoing, mirroring and summarising; NURSE: naming, understanding, respecting, supporting and exploring),[14 15] as well as the following general communication factors rated by the study team: information given regarding diagnosis, symptoms, treatment steps, social issues; talking 'about' instead of 'with' the patient; communicating at cross purposes. Furthermore, moments were identified, when raters had the impression that a patient did not understand information but did not ask more questions; instances when the physician explicitly disagreed with or corrected a patient; when a patient disagreed with/corrected a physician; occurrence of Current Unvoiced Elements (*cues*) and concerns; addressing *cue* and concerns; and replying by providing information instead of exploration. *Cues* and concerns were defined as proposed by previous research.[16] In short, a *cue* or concern included a verbally or non-verbally expressed hint expressed by the patient that might have a certain subjective importance and a negative emotional impact.

### Primary and secondary endpoints

The primary endpoint for this analysis was patients' overall satisfaction, defined by the mean of several satisfaction measures (see secondary endpoints) and measured on a Visual Analogue Scale (VAS) of 0–100, with 0 indicating the lowest and 100 the highest possible satisfaction. For the purpose of this study, patient satisfaction below the median were defined as low satisfaction. Secondary endpoints included satisfaction outcomes for: ward rounds, hospital stay, medical care, physician communication and nursing team communication, measured on the same a VAS from 0 to 100.

We also assessed three elements of patients' subjective and objective knowledge of their own medical situation: the understanding of the disease, therapeutic approach and further plans for care. Each dimension was rated by the patient on a VAS from 0 to 100 (0 'no knowledge about the situation' to 100 'best possible knowledge about the situation'). Subjective knowledge was defined as the patient's self-assessment of being informed and was rated during a structured interview in the afternoon after the ward round. The study team rated objective knowledge by comparing patients' recall of information about their main disease with medical information from the chart.

Ward round duration was measured in minutes and included: total time during the ward round, individual outside the room discussions or bedside discussions and debriefing outside the room.

Further secondary endpoints included multiple items relating to patients' perception of: time spent on the ward round, patients' discomfort during the ward round, physician behaviour during the ward round and general quality of care (online supplemental tables). All were rated in the structured interview after the ward round and measured on a VAS from 0 to 100.

## Statistical analysis

Baseline parameters and outcomes were stratified among patients with and without sensitive topics. Logistic regression analyses were conducted to evaluate factors associated with sensitive topics. Furthermore, univariable linear and logistic regression models were conducted to investigate associations of sensitive topics with primary and secondary endpoints. We additionally calculated multivariable models adjusted for study centre and randomisation arm (bedside or hallway). In the subgroup of patients with sensitive topics, potential risk factors for low satisfaction compared with high satisfaction were assessed using Student's t-test. We used Stata V.15.0 (StataCorp, College Station, Texas, USA) for all statistical analyses. A p value of $<0.05$ (two-tailed) was considered statistically significant. Stata V.15.0 (StataCorp, College Station, Texas, USA) was used for all analyses.

## RESULTS

### Study flow of the original trial

Of 1441 patients approached for inclusion in the original trial, 1092 patients agreed and gave written informed consent. After exclusions, 919 patients were included in the final analysis.[6]

### Baseline characteristics

A total of 474 (51.6%) patients had at least one sensitive topic needing discussion during the ward round. In total, 791 sensitive topics emerged during ward round discussions, including: medical uncertainty (n=251), psychiatric comorbidities (n=161), tumour diagnosis (n=137), social issues (n=125), non-adherence (n=43), previous conflicts between patient and treating team (n=38) and treatment

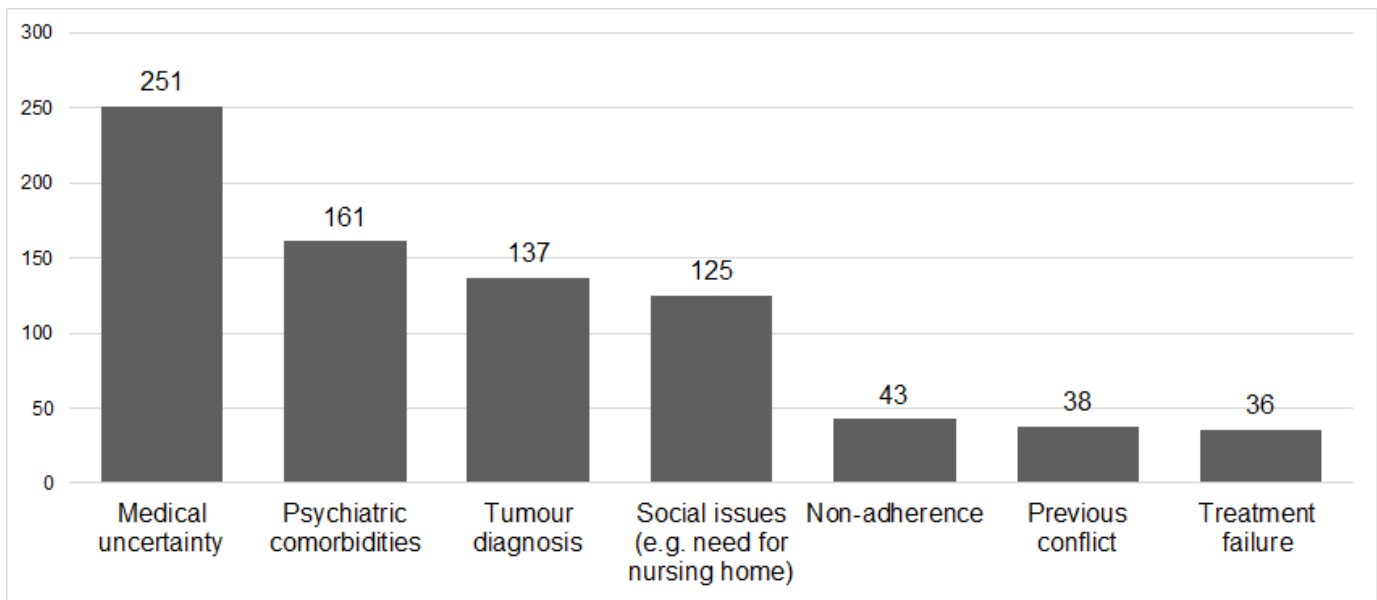

**Figure 1** Sensitive topics and their frequency (n=791).

failure (n=36) (figure 1, table 1 and online supplemental table 1).

The mean age was 65 years±15.9 years (SD). Patients with and without sensitive topics were of comparable age. Overall, 39.3% (n=361) of patients were women with no differences regarding the occurrence of sensitive topics to that of men. Patients' number of children, family status, citizenship, level of education and occupation were not associated with the presence of sensitive topics. Individuals with psychiatric comorbidities, however, were more likely to have sensitive topics present (73/474 (15.4%) vs 31/445 (7.0%), p<0.001; adjusted OR 1.97, 95% CI 1.21 to 3.2; p=0.006). Quality of life regarding mobility, self-care, usual activities, pain/discomfort and anxiety/depression was perceived lower by patients with sensitive topics; consequently resulting in a lower EQ-5D quality of life index score.

### Communication-related factors

Patients with sensitive health topics provided more *cues* (79/474 (16.7%) vs 32/445 (7.2%), p<0.001; adjusted OR 2.36, 95% CI 1.52 to 3.66; p<0.001) and *cues* were more often addressed by physicians in patients with sensitive topics (40/474 (8.4%) vs 19/445 (4.3%), p=0.01, adjusted OR 1.83, 95% CI 1.03 to 3.24; p=0.039). In the subgroup of patients who mentioned emotions, physicians applied patient-centred communication techniques (ie, NURSE, WEMS) similarly among individuals with and without sensitive topics (table 2, online supplemental file 1).

### Primary and secondary outcomes

Patients with sensitive topics reported lower overall satisfaction (87.7±14.6 vs 90.2±12.1, p=0.006; adjusted difference −2.5, 95% CI −4.28 to −0.72; p=0.006). Sensitive topics were associated with less subjective knowledge (77.8±22 vs 81.2±19, p=0.013; adjusted difference −3.86,

95% CI −6.57 to −1.15; p=0.005) as well as objective knowledge (68.7±25.8 vs 72.6±24.9, p=0.021; adjusted difference −3.83, 95% CI −7.18 to −0.48; p=0.025) (table 3).

Duration of outside the room discussions, bedside discussions and debriefings was longer in patients with sensitive topics. Mean (±SD) duration (min) of ward round per patient was 14.5±5.6 versus 11.3±4.6, p<0.001; adjusted difference 3.13, 95% CI 2.47 to 3.79; p<0.001. Compared with patients without sensitive topics, patients with sensitive topics felt more uncomfortable (4.5±15.6 vs 8.2±21.6, p=0.005; adjusted difference 3.2, 95% CI 0.64 to 5.76; p=0.014) and unsettled (4.4±15.9 vs 7.2±19.9, p=0.033; adjusted difference 2.84, 95% CI 0.22 to 5.45; p=0.034) during the ward round, and discussions causing them to worry (6.6±25.7 vs 11.7±24.8, p=0.003; adjusted difference 4.82, 95% CI 1.4 to 8.24; p=0.006). They also felt less confident with the physician team (91.9±14.4 vs 89.5±16.8, p=0.02; adjusted difference −2.52, 95% CI −4.65 to −0.4; p=0.02), whereas there were no significant differences in confidence with the nursing team (online supplemental table 3).

Patients with sensitive topics perceived physicians less compassionate and also felt less respected, taken seriously, and more uneasy (table 3).

### Risk factors for low satisfaction among patients with sensitive topics

In a further step, we investigated factors associated with low satisfaction among patients with sensitive topics (table 4). Several factors were associated with low satisfaction, such as parameters regarding the patient–physician interaction during ward rounds such as physicians disagreeing with patients (14/212 (6.6%) vs 41/254 (16.1%), adjusted OR 2.78, 95% CI 1.47 to 5.27; p=0.002) and patients disagreeing with physicians (14/212 (6.6%) vs 37/254 (14.6%), adjusted OR 2.42, 95% CI 1.27 to 4.61; p=0.007). Further factors were

**Table 1** Associations of patient characteristics with sensitive topics

| | n | All | No sensitive topics | Sensitive topics | P value | OR (95% CI)* | P value |
|---|---|---|---|---|---|---|---|
| n | | | 445 | 474 | | | |
| **Socio-demographic factors** | | | | | | | |
| Age, years; mean (SD) | 919 | 65.0 (15.9) | 65.7 (15.9) | 64.3 (15.9) | 0.16 | 1 (0.99 to 1) | 0.304 |
| Female sex, n (%) | 919 | 361 (39.3) | 173 (38.9) | 188 (39.7) | 0.81 | 1.05 (0.8 to 1.37) | 0.741 |
| Number of children, mean (SD) | 919 | 2.5 (9.2) | 2.1 (6.7) | 2.8 (11.0) | 0.26 | 1.01 (0.99 to 1.02) | 0.365 |
| Family status | 919 | | | | | | |
| Single, relationship, married, registered partnership, n (%) | | 648 (70.5) | 317 (71.2) | 331 (69.8) | 0.84 | | |
| Separated, divorced, n (%) | | 147 (16.0) | 68 (15.3) | 79 (16.7) | | 1.05 (0.73 to 1.52) | 0.787 |
| Widowed, n (%) | | 124 (13.5) | 60 (13.5) | 64 (13.5) | | 1.07 (0.72 to 1.59) | 0.738 |
| Citizenship | 919 | | | | | | |
| Switzerland, n (%) | | 789 (85.9) | 386 (86.7) | 403 (85.0) | 0.71 | | |
| Germany, n (%) | | 55 (6.0) | 26 (5.8) | 29 (6.1) | | 0.97 (0.55 to 1.68) | 0.904 |
| Other, n (%) | | 75 (8.2) | 33 (7.4) | 42 (8.9) | | 1.22 (0.75 to 1.98) | 0.421 |
| Occupation | 919 | | | | | | |
| Employed/working+IV, n (%) | | 259 (28.2) | 129 (29.0) | 130 (27.4) | **0.041** | | |
| Unemployed/homemaker, n (%) | | 39 (4.2) | 13 (2.9) | 26 (5.5) | | 2 (0.97 to 4.09) | 0.059 |
| Retired/IV support, n (%) | | 605 (65.8) | 296 (66.5) | 309 (65.2) | | 1.05 (0.78 to 1.42) | 0.735 |
| In education, n (%) | | 6 (0.7) | 5 (1.1) | 1 (0.2) | | 0.19 (0.02 to 1.63) | 0.128 |
| Other, n (%) | | 10 (1.1) | 2 (0.4) | 8 (1.7) | | 3.41 (0.7 to 16.56) | 0.128 |
| Comorbidities | 919 | | | | | | |
| Charlson Comorbidity Index, mean (SD) | | 4.45 (2.88) | 4.4 (2.8) | 4.5 (3.0) | 0.54 | 0.76 (0.59 to 1) | **0.048** |
| Cardiology, n (%) | | 485 (52.8) | 248 (55.7) | 237 (50.0) | 0.082 | 0.86 (0.65 to 1.13) | 0.273 |
| Neurology, n (%) | | 193 (21.0) | 80 (18.0) | 113 (23.8) | **0.029** | 1.01 (0.77 to 1.33) | 0.941 |
| Rheumatology/Immunology, n (%) | | 143 (15.6) | 68 (15.3) | 75 (15.8) | 0.82 | 1.03 (0.78 to 1.37) | 0.819 |
| Gastrointestinal, n (%) | | 254 (27.6) | 118 (26.5) | 136 (28.7) | 0.46 | 1.18 (0.85 to 1.65) | 0.325 |
| Endocrinology, n (%) | | 352 (38.3) | 185 (41.6) | 167 (35.2) | **0.048** | 1.21 (0.86 to 1.71) | 0.264 |
| Respiratory, n (%) | | 315 (34.3) | 152 (34.2) | 163 (34.4) | 0.94 | 1.07 (0.81 to 1.4) | 0.627 |
| Infectious diseases, n (%) | | 171 (18.6) | 77 (17.3) | 94 (19.8) | 0.33 | 1.12 (0.85 to 1.48) | 0.416 |
| Renal, n (%) | | 317 (34.5) | 150 (33.7) | 167 (35.2) | 0.63 | 0.25 (0.07 to 0.91) | **0.035** |
| Gynaecology, n (%) | | 14 (1.5) | 11 (2.5) | 3 (0.6) | **0.023** | 0.26 (0.07 to 0.94) | **0.04** |
| Urology, n (%) | | 93 (10.1) | 43 (9.7) | 50 (10.5) | 0.66 | 1.24 (0.8 to 1.93) | 0.338 |
| Oncology, n (%) | | 287 (31.2) | 145 (32.6) | 142 (30.0) | 0.39 | 1.81 (1.13 to 2.9) | **0.014** |
| Psychiatry, n (%) | | 104 (11.3) | 31 (7.0) | 73 (15.4) | **<0.001** | 1.97 (1.21 to 3.2) | **0.006** |
| Depression, n (%) | | 82 (8.9) | 29 (6.5) | 53 (11.2) | **0.013** | 0.89 (0.68 to 1.15) | 0.364 |
| Other, n (%) | | 498 (54.2) | 248 (55.7) | 250 (52.7) | 0.36 | 0.95 (0.73 to 1.25) | 0.734 |
| Health self-rating VAS (0–100), mean (SD) | 856 | 56.9 (23.0) | 58.7 (22.5) | 55.1 (23.3) | **0.021** | 0.99 (0.99 to 1) | **0.018** |
| Quality of life (European Quality of Life 5 Dimensions 3 Level Version) index, mean (SD) | 889 | | 0.749 (0.278) | 0.671 (0.301) | **<0.001** | 0.42 (0.26 to 0.67) | **<0.001** |

Significant p-values were marked in bold.
*Adjusted for study centre, intervention. ORs were calculated with logistic regression models.
IV, Swiss disability insurance; n, number; VAS, Visual Analogue Scale.

a patient's lack of knowledge about their own medical care (both subjective and objective knowledge), an overall longer duration of the ward round, reduced capacity to understand the main disease, the implemented therapeutic measure and further plans of care, as well as other physician–patient interactions (eg, respectful treatment, physician compassion, observing privacy), see also table 4, online supplemental table 4.

**Table 2** Associations of communication factors with sensitive topics

| | n | No sensitive topics | Sensitive topics | P value | Adjusted OR (95% CI)* | P value |
|---|---|---|---|---|---|---|
| | | 445 | 474 | | | |
| Communication factors | 919 | | | | | |
| Visible evidence of emotion, n (%) | | 48 (10.8) | 70 (14.8) | 0.071 | 1.35 (0.9 to 2.01) | 0.144 |
| Medical staff talks to patient about diagnosis, n (%) | | 215 (48.3) | 271 (57.2) | **0.007** | 1.59 (1.21 to 2.09) | **0.001** |
| Medical staff talks to patient about symptoms, n (%) | | 399 (89.7) | 443 (93.5) | **0.038** | 1.66 (1.02 to 2.69) | **0.041** |
| Medical staff talks to patient about treatment, n (%) | | 334 (75.1) | 383 (80.8) | **0.036** | 1.38 (1 to 1.9) | **0.049** |
| Medical staff talks to patient about next steps, n (%) | | 344 (77.3) | 431 (90.9) | **<0.001** | 2.8 (1.89 to 4.13) | **<0.001** |
| Medical staff talks to patient about social issues, n (%) | | 272 (61.1) | 320 (67.5) | **0.043** | 1.43 (1.08 to 1.9) | **0.013** |
| Medical staff talks about patient instead of with patient after case discussion, n (%) | | 329 (73.9) | 351 (74.1) | 0.97 | 1.19 (0.87 to 1.62) | 0.282 |
| Physician and patient talk at cross purposes, n (%) | | 17 (3.8) | 35 (7.4) | **0.019** | 2.29 (1.24 to 4.25) | **0.008** |
| Patient seems not to understand something, but does not ask, n (%) | | 6 (1.3) | 17 (3.6) | **0.030** | 4.15 (1.55 to 11.08) | **0.005** |
| Physician disagrees with/corrects patient, n (%) | | 30 (6.7) | 57 (12.0) | **0.006** | 2.05 (1.28 to 3.3) | **0.003** |
| Patient disagrees with/corrects physician, n (%) | | 27 (6.1) | 53 (11.2) | **0.006** | 2.12 (1.29 to 3.48) | **0.003** |
| Occurrence of 'cue', n (%) | | 32 (7.2) | 79 (16.7) | **<0.001** | 2.36 (1.52 to 3.66) | **<0.001** |
| Addressing 'cue' concern, n (%) | | 19 (4.3) | 40 (8.4) | **0.010** | 1.83 (1.03 to 3.24) | **0.039** |
| Reacts by providing information instead of exploration, n (%) | | 19 (4.3) | 33 (7.0) | 0.077 | 1.59 (0.88 to 2.87) | 0.122 |

Significant p-values were marked in bold.
*Adjusted for study centre, intervention. ORs were calculated with logistic regression models.
cue, Current Unvoiced Element; n, number.

## Qualitative analysis

Examples of conversations extracted from audiotapes are provided in online supplemental table 5.

## DISCUSSION

We report several key findings of this ancillary project from a multicentre randomised controlled trial that investigated sensitive topics and specific elements recorded during ward round communication: (1) Sensitive topics arose frequently during ward rounds, with more than half of patients reporting at least one specific issue. Participants reporting a sensitive topic were more complex in their treatment and required more attention from medical staff, making ward rounds longer and more demanding compared with individuals without a sensitive topic; (2) medical uncertainty was the most frequent sensitive topic; (3) the presence of sensitive topics in patients was associated with being less satisfied overall with their care; (4) risk factors for low satisfaction among patients with sensitive topics included several parameters regarding patient–physician interaction during the ward round (various disagreements)—highlighting the importance of patient-centred communication.

Several results of the analysis provide important information and are worth further discussion. Many issues could benefit from special attention, but often take time and may not fit into the ward round time-schedule. However, it is important to address them, as patients may feel unheard and dissatisfied if these topics are not discussed (see an example of a social issue from qualitative analysis in box 1).

Another frequently encountered sensitive topic concerned tumour diagnosis. Previous literature has shown an association between physicians' communicative competence and its effect on quality of life in patients with cancer.[17] As quality of life was also lower in patients with sensitive topics in our study, this again underlines the importance of communication skills when talking about tumour diagnoses.

Psychiatric comorbidities and psychological issues are also perceived as sensitive topics and may be directly linked to physical disease. Physicians often hesitate to address mental health issues during the ward round because of a lack of privacy.[7] However, our results suggest that patients do not feel violated in their privacy when sensitive topics are addressed (see online supplemental table 3). Also, previous research has shown that 63% of patients prefer to discuss mental health issues in general, at least in the primary care setting.[18] However, similar data for the inpatient setting with less privacy is currently lacking.

**Table 3** Association of sensitive topics with various outcomes

| | n | No sensitive topics | Sensitive topics | P value | Adjusted difference or OR (95% CI)* | P value |
|---|---|---|---|---|---|---|
| Primary endpoint | | 445 | 474 | | | |
| Overall satisfaction (VAS 0–100), mean (SD) | 906 | 90.2 (12.1) | 87.7 (14.6) | **0.006** | −2.5 (−4.28 to 0.72) | **0.006** |
| Patient knowledge about medical care | | | | | | |
| Average subjective knowledge about their medical care, mean (SD) | 919 | 81.2 (19.0) | 77.8 (22.0) | **0.013** | −3.86 (−6.57 to 1.15) | **0.005** |
| Average objective knowledge about their medical care, mean (SD) | 919 | 72.6 (24.9) | 68.7 (25.8) | **0.021** | −3.83 (−7.18 to 0.48) | **0.025** |
| Measured timeliness of ward round | | | | | | |
| Duration of outside the room discussions (min), mean (SD) | 919 | 2.8 (3.6) | 4.9 (5.0) | **<0.001** | 1.41 (1.02 to 1.81) | **<0.001** |
| Duration of bedside discussions (min), mean (SD) | 919 | 8.2 (4.2) | 9.0 (5.0) | **0.005** | 1.56 (1.03 to 2.09) | **<0.001** |
| Duration of debriefing outside the room (min), mean (SD) | 919 | 0.4 (1.1) | 0.5 (1.1) | 0.063 | 0.16 (0.01 to 0.31) | **0.038** |
| Total duration of ward round per patient (min), mean (SD) | 919 | 11.3 (4.6) | 14.5 (5.6) | **<0.001** | 3.13 (2.47 to 3.79) | **<0.001** |
| Patient perception regarding discomfort during the ward round | | | | | | |
| Medical terms used during ward round were confusing (VAS 0–100), mean (SD) | 853 | 15.9 (28.2) | 19.2 (30.4) | 0.099 | 3.89 (−0.07 to 7.85) | 0.054 |
| Ward round discussions made me worry (VAS 0–100), mean (SD) | 871 | 6.6 (25.7) | 11.7 (24.8) | **0.003** | 4.82 (1.4 to 8.24) | **0.006** |
| I felt uncomfortable during ward round (VAS 0–100), mean (SD) | 870 | 4.5 (15.6) | 8.2 (21.6) | **0.005** | 3.2 (0.64 to 5.76) | **0.014** |
| Ward round discussions unsettled me (VAS 0–100), mean (SD) | 752 | 4.4 (15.9) | 7.2 (19.9) | **0.033** | 2.84 (0.22 to 5.45) | **0.034** |
| Patient perception regarding physician' behaviour during the ward round | | | | | | |
| Physicians treated me with respect (VAS 0–100), mean (SD) | 867 | 97.4 (8.8) | 95.5 (13.1) | **0.011** | −2.05 (−3.57 to −0.53) | **0.008** |
| I was taken seriously (VAS 0–100), mean (SD) | 861 | 96.1 (10.9) | 94.0 (15.6) | **0.021** | −2.17 (−4.01 to −0.32) | **0.021** |
| Physicians respected my privacy (VAS 0–100), mean (SD) | 794 | 89.7 (24.8) | 91.1 (20.9) | 0.38 | 1.41 (−1.85 to 4.66) | 0.396 |
| Physicians showed compassion (VAS 0–100), mean (SD) | 693 | 81.0 (28.4) | 75.2 (33.5) | **0.014** | −5.56 (−10.33 to 0.8) | **0.022** |
| My issues were dealt with discreetly (VAS 0–100), mean (SD) | 764 | 88.3 (23.7) | 89.2 (21.1) | 0.55 | 1.33 (−1.91 to 4.57) | 0.42 |
| Some topics during ward round communication caused inconvenience (VAS 0–100), mean (SD) | 853 | 4.0 (15.4) | 7.6 (21.8) | **0.005** | 3.58 (0.99 to 6.17) | **0.007** |

Significant p-values were marked in bold.
*Adjusted for study centre, intervention. All differences calculated with linear regression models for continuous data. ORs were calculated with logistic regression models.
min, minutes; VAS, Visual Analogue Scale.

Recent studies highlight that medical uncertainty remains a significant factor in medicine despite the advances in diagnostic possibilities particularly in patients reporting various unspecific symptoms.[19] Thus, during ward rounds, the treating team often discusses possible differential diagnoses leaving the patient potentially with remaining uncertainty about the specific disease. This is in line with our results where medical uncertainty was reported as the most common sensitive topic. Medical uncertainty has been shown to negatively affect patients' physical, and mental, well-being.[20] Moreover, dealing with medical uncertainty may also cause anxiety in physicians, negatively affect their work-related satisfaction and result in substandard care.[21] Guidance on how best to deal with medical uncertainty and how to address it in the patient's presence is warranted to ensure both physician well-being and patient-centred care.

**Table 4** Risk factors for low satisfaction among patients with sensitive topics

| | n | High satisfaction | Low satisfaction | P value | Adjusted difference or OR (95% CI)* | P value |
|---|---|---|---|---|---|---|
| | | 212 | 254 | | | |
| Other communication factors | 466 | | | | | |
| Physician disagrees with/corrects patient, n (%) | | 14 (6.6) | 41 (16.1) | 0.001 | 2.78 (1.47 to 5.27) | 0.002 |
| Patient disagrees with/corrects physician, n (%) | | 14 (6.6) | 37 (14.6) | 0.006 | 2.42 (1.27 to 4.61) | 0.007 |
| Patient knowledge about medical care | 466 | | | | | |
| Average subjective knowledge about their medical care, mean (SD) | | 87.0 (16.2) | 70.7 (22.9) | <0.001 | 0.95 (0.94 to 0.97) | <0.001 |
| Average objective knowledge about their medical care, mean (SD) | | 71.7 (25.3) | 66.2 (26.0) | 0.023 | 0.99 (0.98 to 1) | 0.027 |
| Measured timeliness of ward round | 466 | | | | | |
| Total duration of ward round per patient (min), mean (SD) | | 13.9 (5.6) | 15.1 (5.6) | 0.019 | 1.05 (1.01 to 1.08) | 0.01 |
| Patient perception regarding time spent on ward round | 466 | | | | | |
| Overall duration of ward round was sufficient (VAS 0–100), mean (SD) | | 95.78 (15.38) | 83.81 (24.55) | <0.001 | 0.96 (0.95 to 0.98) | <0.001 |
| Time spent with physicians was sufficient (VAS 0–100), mean (SD) | | 96.71 (10.02) | 80.66 (22.53) | <0.001 | 0.92 (0.9 to 0.94) | <0.001 |
| Patient estimation of time spent with patient on ward round (min), mean (SD) | | 12.18 (7.18) | 11.26 (6.58) | 0.15 | 0.98 (0.96 to 1.01) | 0.176 |
| Patient estimation of time spent per day with patient case overall (min), mean (SD) | | 82.09 (86.72) | 66.73 (66.44) | 0.031 | 1 (0.99 to 1) | 0.035 |
| The ward round was helpful for better | | | | | | |
| Understanding the main illness (VAS 0–100), mean (SD) | | 75.03 (32.91) | 54.64 (33.95) | <0.001 | 0.98 (0.98 to 0.99) | <0.001 |
| Further therapeutic measures (VAS 0–100), mean (SD) | | 74.14 (32.45) | 53.03 (33.52) | <0.001 | 0.98 (0.97 to 0.99) | <0.001 |
| Further plans of care (VAS 0–100), mean (SD) | | 83.33 (69.99) | 62.12 (52.73) | <0.001 | 0.99 (0.98 to 0.99) | <0.001 |
| All my questions were answered (VAS 0–100), mean (SD) | | 97.05 (8.55) | 82.10 (24.16) | <0.001 | 0.92 (0.9 to 0.94) | <0.001 |
| I was able to understand all answers to my questions (VAS 0–100), mean (SD) | | 96.84 (10.74) | 85.85 (20.14) | <0.001 | 0.93 (0.91 to 0.95) | <0.001 |
| Information during visit has been adequate, n (%) | | 197 (92.9) | 176 (69.3) | <0.001 | 0.17 (0.1 to 0.31) | <0.001 |
| Patient did not understand something during the round, n (%) | | 13 (6.1) | 36 (14.2) | 0.005 | 2.58 (1.32 to 5.02) | 0.005 |
| Estimation of my participation in the discussion (VAS 0–100), mean (SD) | | 67.80 (30.61) | 51.78 (29.76) | <0.001 | 0.98 (0.98 to 0.99) | <0.001 |
| Disease was explained in an understandable way (VAS 0–100), mean (SD) | | 91.95 (19.46) | 76.61 (27.1) | <0.001 | 0.97 (0.96 to 0.98) | <0.001 |
| Treatment was explained in an understandable way (VAS 0–100), mean (SD) | | 91.32 (20.51) | 69.49 (31.19) | <0.001 | 0.96 (0.95 to 0.97) | <0.001 |
| Upcoming examinations were explained in an understandable way (VAS 0–100), mean (SD) | | 91.64 (19.12) | 74.34 (28.26) | <0.001 | 0.96 (0.95 to 0.97) | <0.001 |
| The information given during the visit was clear and understandable (VAS 0–100), mean (SD) | | 97.86 (9.04) | 82.92 (22.92) | <0.001 | 0.9 (0.88 to 0.93) | <0.001 |
| Patient perception regarding discomfort during the ward round | 466 | | | | | |

Continued

**Table 4** Continued

| | n | High satisfaction | Low satisfaction | P value | Adjusted difference or OR (95% CI)* | P value |
|---|---|---|---|---|---|---|
| Medical terms used during ward round were confusing (VAS 0–100), mean (SD) | | 14.1 (27.87) | 23.13 (30.42) | **0.003** | 1.01 (1 to 1.02) | **0.003** |
| Ward round discussions made me worry (VAS 0–100), mean (SD) | | 7.93 (22.02) | 15.86 (25.73) | **<0.001** | 1.02 (1.01 to 1.02) | **0.001** |
| I felt uncomfortable during ward round (VAS 0–100), mean (SD) | | 3.8 (14.88) | 11.98 (24.44) | **<0.001** | 1.02 (1.01 to 1.04) | **<0.001** |
| Ward round discussions unsettled me (VAS 0–100), mean (SD) | | 2.23 (9.31) | 12.01 (22.3) | **<0.001** | 1.06 (1.03 to 1.08) | **<0.001** |
| Patient perception regarding physician's behaviour during the ward round | | | | | | |
| Physicians treated me with respect (VAS 0–100), mean (SD) | 466 | 99.31 (3.02) | 91.87 (16.37) | **<0.001** | 0.85 (0.8 to 0.9) | **<0.001** |
| I was taken seriously (VAS 0–100), mean (SD) | 466 | 98.82 (4.97) | 88.87 (19.54) | **<0.001** | 0.89 (0.85 to 0.92) | **<0.001** |
| Physicians respected my privacy (VAS 0–100), mean (SD) | 466 | 95.83 (14.64) | 86.64 (21.55) | **<0.001** | 0.96 (0.94 to 0.98) | **<0.001** |
| Physicians showed compassion (VAS 0–100), mean (SD) | 466 | 83.98 (27.57) | 66.79 (30.85) | **<0.001** | 0.98 (0.97 to 0.99) | **<0.001** |
| My issues were dealt with discreetly (VAS 0–100), mean (SD) | 466 | 95.01 (14.75) | 83.95 (21.16) | **<0.001** | 0.95 (0.94 to 0.97) | **<0.001** |
| Some topics during ward round communication caused inconvenience (VAS 0–100), mean (SD) | 466 | 3.66 (15.99) | 10.88 (23.80) | **<0.001** | 1.02 (1.01 to 1.03) | **0.001** |
| I was encouraged to address personal topics (VAS 0–100), mean (SD) | 466 | 90.57 (21.35) | 76.54 (27.65) | **<0.001** | 0.97 (0.96 to 0.98) | **<0.001** |
| My privacy was violated (VAS 0–100), mean (SD) | 466 | 0.88 (5.30) | 5.19 (15.55) | **<0.001** | 1.07 (1.02 to 1.12) | **0.003** |
| Teaching took place during ward round (yes, %) | 466 | 61 (28.8) | 65 (25.6) | 0.44 | 0.87 (0.57 to 1.33) | 0.51 |
| If yes, teaching was perceived as disruptive, (VAS 0–100) mean (SD) | 125 | 0.8 (6.4) | 10.3 (29.3) | **0.015** | 1.04 (1 to 1.08) | 0.07 |
| Patients' perception regarding quality of care (VAS 0–100) | 466 | | | | | |
| I felt 'in good hands' in this hospital, mean (SD) | | 96.19 (9.28) | 83.57 (17.04) | **<0.001** | 0.9 (0.88 to 0.93) | **<0.001** |
| I felt there were contradicting statements from physicians and nursing team, mean (SD) | | 7.58 (20.90) | 18.87 (28.23) | **<0.001** | 1.02 (1.01 to 1.03) | **<0.001** |
| I feel confident with the physician team, mean (SD) | | 95.88 (11.54) | 83.79 (18.07) | **<0.001** | 0.92 (0.9 to 0.94) | **<0.001** |
| I feel confident with the nursing team, mean (SD) | | 97.45 (6.46) | 86.91 (15.65) | **<0.001** | 0.89 (0.87 to 0.92) | **<0.001** |
| Physicians and nurses collaborate well, mean (SD) | | 95.68 (9.14) | 84.25 (15.8) | **<0.001** | 0.91 (0.89 to 0.93) | **<0.001** |
| I feel physicians are highly competent to treat the current illness, mean (SD) | | 96.35 (9.71) | 82.84 (18.49) | **<0.001** | 0.9 (0.88 to 0.93) | **<0.001** |
| I feel nurses are highly competent to treat the current illness, mean (SD) | | 96.51 (9.16) | 86.39 (31.79) | **<0.001** | 0.93 (0.91 to 0.95) | **<0.001** |

Significant p-values were marked in bold.
*Adjusted for centrum, intervention. All differences calculated with linear regression models for continuous data. ORs were calculated with logistic regression models.
min, minutes; VAS, Visual Analogue Scale.

Specific communication skills training may help achieve these aims.[19 22]

Importantly, our study found lower satisfaction in patients with sensitive topics compared with patients without sensitive topics. This may be due to confounding because patients with sensitive topics are often more complex and have different medical needs. Interestingly, previous literature in outpatient settings suggests that adequately addressing sensitive topics is associated with higher satisfaction.[8 9] Santelli et al reported higher levels

**Box 1  Example of a patient–physician discussion about social burden**

Patient: I planned to move in with my partner after my hospital stay. She's a former nurse, but now a relative of hers has died.
Chief physician: Oh dear.
Patient (tearful): When she heard what could happen to me, she said no. I mean, over the last years I've burdened her with my illness. But now I am really down.
Chief physician: Yes, that is… I can understand that.
Patient: Now I had to agree with her and promised to look for assisted accommodation.
Chief physician: Yes, we can discuss this with our case management team. Or have you already found something?
Patient: No, because I expected to move in with her. But now her relative has passed away and it is all so complicated…
Chief physician: Yeah, I understand! But unfortunately, we don't have much time during the ward round so we'll have to discuss your further care on another occasion.
Patient: Okay, yes, thank you.

**Box 2  Example of a disagreement between patient and physician**

Chief physician: We know that you normally take Diazepam, which has a long half-life…
Patient: Yes, 24 hours.
Chief physician: Even longer, especially if the liver is not working properly. I've seen two people die from Diazepam. We prefer to prescribe Lorazepam.
Patient: I do not tolerate Lorazepam.
Chief physician: What do you mean by "don't tolerate"?
Patient: Once when I withdrew from alcohol and doctors gave me Lorazepam I hallucinated. I saw a mouse in my room. Then I had delirium and woke up in another hospital.
Chief physician: We cannot assume that this was because of the Lorazepam. It was more likely due to alcohol withdrawal.
Patient: Yes, but I've never experienced anything like that with Diazepam.
Chief physician: It is important to me that we informed you about this…

of confidentiality during the conversation when sensitive topics were discussed.[23] We did not have information about the extent to which the sensitive topics were discussed with patients in our study. The restricted opportunity for confidentiality during ward rounds may provide some barriers here—even though patients did not expressly state that their privacy was violated. Limited time with the physician may be another contributor to the discrepancy in satisfaction between outpatient and inpatient settings. Especially time constraints during ward rounds might be a challenge to effectively address sensitive topics, compromising patient satisfaction. However, most patients stated that the time was sufficient, and the duration of ward rounds was significantly longer in patients with sensitive topics. Also, we also found the duration of ward rounds to be associated with lower satisfaction in patients with sensitive topics. It would be important to study the effect of addressing sensitive topics in the inpatient population regarding satisfaction with care in an interventional study.

Over the last decades, medicine has shifted from a paternalistic to a participatory, patient-involving model, with shared decision-making as a key element. Schifferli hypothesises that disagreements between patients and physicians may be a consequence of increasing patient involvement and highlights the importance of patients' wish for participation in clinical decision-making.[24] However, not all patients may prefer shared decision-making and a personalised approach in this regard may be needed.[25] We found that disagreements between physicians and patients were a significant risk factor for low satisfaction in patients with sensitive topics. The example of our qualitative analysis in box 2 shows a potential for disagreement when patients wish to actively participate in the definition of treatment options, and the concept of best care differs between the patient and physician.

A recent study showed that patients with active decisional preference are less satisfied with their care and have less trust in the healthcare team compared with patients preferring less in depth involvement in medical decisions.[25] This is in line with the present findings, suggesting that disagreements are a risk factor for low satisfaction. A patient's wish for participation is therefore relevant, and a more personalised approach may improve the patient–physician relationship and increase patients' satisfaction with medical care. However, as shown in our study, correctly addressing sensitive topics remains a major challenge during ward rounds due to several factors including lack of privacy, time constraints, lack of training and standards on how to best address sensitive topics among others. Future research and quality initiatives should focus on this important issue to improve the care of patients with sensitive topics.

This trial has several limitations. First, we only included Swiss teaching hospitals. This might cause a selection bias and limits generalisability of the findings. Second, using a pragmatic approach, ward rounds in the three participating hospitals were not standardised regarding clinical practice and to ensure external validity. Consequently, the internal validity might be reduced. Third, the questionnaire used to assess patient perception has not yet been validated. Finally, there is no well accepted definition of sensitive topics and we thus defined it based on the clinical experience of the physician–researcher team. Still, there is some data suggesting that patients find value and comfort when their doctors openly discuss uncertainty with them.[26] Yet, a sensitivity analysis excluding uncertainty from the definition of sensitive topics showed similar results.

## Conclusion

This analysis suggests that a large proportion of a broad medical inpatient sample have sensitive health topics, which was associated with lower satisfaction with care—particularly if the patient perceived patient–physician interactions during ward rounds as unsatisfactory. Specific physicians training in communication techniques how to

identify, discuss and address sensitive topics may help to improve the care of these patients. Besides improving communication skills, future studies need to address the question which sensitive topics can be discussed at the bedside or in a more private setting.

**Author affiliations**
¹Department of Medical Communication / Psychosomatic Medicine, University Hospital Basel, Basel, Switzerland
²Division of Clinical Psychology and Psychotherapy, Faculty of Psychology, University of Basel, Basel, Switzerland
³Emergency Department, University Hospital Basel, Basel, Switzerland
⁴Faculty of Medicine, University of Basel, Basel, Switzerland
⁵Division of Internal Medicine, Kantonsspital Aarau, Aarau, Switzerland
⁶University Center of Internal Medicine, Kantonsspital Baselland, Liestal, Switzerland
⁷Department of Psychosomatics and Psychiatry, Bethesda Hospital, Basel, Switzerland
⁸Department of Internal Medicine, University Hospital Basel, Basel, Switzerland

**Contributors** All listed authors were involved in conducting the present study. SG designed, analysed and interpreted this ancillary analysis, and drafted the article. CB designed the original trial, wrote the proposal for the ethics committee, provided study material or patients and critically revised the manuscript for important intellectual content. KB collected patient data and critically revised the manuscript for important intellectual content. VM analysed and interpreted patient data and critically revised the manuscript for important intellectual content. JG contributed to design of this ancillary analyses and critically revised the manuscript for important intellectual content. PS designed the original trial, contributed statistical expertise and critically revised the manuscript for important intellectual content. JDL provided study material or patients and critically revised the manuscript for important intellectual content. RS contributed to the study design and critically revised the manuscript for important intellectual content. WL designed the original trial and critically revised the manuscript for important intellectual content. MT provided study material or patients and critically revised the manuscript for important intellectual content. TB provided study material or patients and critically revised the manuscript for important intellectual content. JE provided study material or patients and critically revised the manuscript for important intellectual content. MO provided study material or patients and critically revised the manuscript for important intellectual content. SB designed the original trial, provided study material or patients and critically revised the manuscript for important intellectual content. SH acts as the author responsible for the overall content as the guarantor, accepts full responsibility for the work and/or the conduct of the study, had access to the data, and controlled the decision to publish, designed the original trial, wrote the proposal for the ethics committee, analysed and interpreted patient data, drafted the article, critically revised the manuscript for important intellectual content, provided study material or patients, contributed statistical expertise and obtained funding. All authors approved the final manuscript.

**Funding** The original trial was funded by the Swiss National Science Foundation (SNSF, REF 10531C_182422).

**Competing interests** SH and her research team are supported by the Swiss National Science Foundation (SNSF) (Ref 10001C_192850/1 and 10531C_182422) and the Gottfried Julia Bangerter-Rhyner Foundation (8472/HEG-DSV). PS has received support from the SNF (SNSF Professorship, PP00P3_150531), the Research Committee (Forschungsrat) of the Kantonsspital Aarau (1410.000.058 and 1410.000.044) and Funds of the Argovia Professorship of the Medical University Clinic (FG 1500000083). PS has previously received unrestricted grant money unrelated to this project from Nestlé Health Science and Abbott Nutrition. JDL and his research team are supported by the Swiss Personalized Health Network (Ref Driver-Project – 2018DR108)) and the Swiss National Science Foundation (SNSF) (Ref SNFS-160072 und -185592). JDL has also received unrestricted grant money unrelated to the project from AstraZeneca AG Switzerland, Boehringer GmbH Switzerland, GSK AG Switzerland and Novartis AG Switzerland. RS received funding from the Stanley Thomas Johnson Stiftung & Gottfried und Julia Bangerter-Rhyner-Stiftung under projects no. PC 28/17 and PC 05/18, from the Swiss Cancer League (Krebsliga Schweiz) under project no. KLS-4304-08-2017, from Promotion Santé Suisse (Gesundheitsförderung Schweiz) under contract no. 18.191/ K50001, and in the context of a Horizon Europe project from the Swiss State Secretariat for Education, Research and Innovation (SERI) under contract number 22.00094. RS received a speaker honorarium from Novartis. MT is recipient of a project grant

of the Swiss National Science Foundation (grant No. 320030_200423) and has research collaborations with Roche, Novartis and Idorsia (all Switzerland).

**Patient and public involvement** Patients and/or the public were involved in the design, or conduct, or reporting, or dissemination plans of this research. Refer to the Methods section for further details.

**Patient consent for publication** Not applicable.

**Ethics approval** The BEDSIDE-OUTSIDE trial including this preplanned ancillary analysis was registered prior to initiation at ClinicalTrials.gov on 7 July 2017 (https://clinicaltrials.gov/ct2/show/NCT03210987) and approved by the local Ethics Committee (Northwest and Central Switzerland, EKNZ, 2017-00991).

**Provenance and peer review** Not commissioned; externally peer reviewed.

**Data availability statement** Data are available upon reasonable request.

**ORCID iDs**
Christoph Becker http://orcid.org/0000-0001-6039-4003
Wolf Langewitz http://orcid.org/0000-0001-8298-7328
Sabina Hunziker http://orcid.org/0000-0003-3648-3609

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
