## [Reviewer comments · BMJ Open]

ARTICLE DETAILS

TITLE (PROVISIONAL)	The occurrence of sensitive topics during ward round: An ancillary analysis of the BEDSIDE-OUTSIDE Trial
AUTHORS	Gross, Sebastian; Becker, Christoph; Beck, Katharina; Memma, Valentina; Gaab, Jens; Schütz, Philipp; Leuppi, Jörg; Schaefert, Rainer; Langewitz, Wolf Axel; Trendelenburg, Marten; Breidhardt, Tobias; Eckstein, J; Osthoff, Michael; Bassetti, Stefano; Hunziker, Sabina

VERSION 1 – REVIEW

REVIEWER	Eid, Thurayya King Saud University
REVIEW RETURNED	22-May-2023

GENERAL COMMENTS	Minor revision page 6 line 59 who approached the patients during recruitment page 7 line 54 the "round" not "Rond" page 7 line 53 name the manufactures of. the voice recording software page 7 line 45 at what time the ward rounds usually occurs. in Australia for example the round time occurs from 09:00 - 11:00 am page 9 line 3 the word "instances" Page 24 line 51 " ward round" page 24 line 53 "differential"
--

REVIEWER	Ratelle, John Mayo Clinic
REVIEW RETURNED	09-Jun-2023

GENERAL COMMENTS	Thank you for the opportunity to review this manuscript. I commend the authors for exploring the important and patient-centered topic of sensitive issues raised during hospital ward rounds. However, I believe there are serious conceptual and methodological issues that threaten the validity of their findings. Major comment: The authors' literature review is insufficient. At several points in the manuscript, they highlight the novelty of their study, the first to "analyze communication regarding sensitive topics during ward rounds.". This claim is categorically false. For example, Jason Satterfield and colleagues studied sensitive topics during ward rounds nearly a decade ago (ref: Acad Med. 2014 Nov;89(11):1548-57. PMID 25250757) Major comment: The authors have not provided an adequate conceptual or operational definition of "sensitive" topics. For example, medical uncertainty and patient non-adherence are
---

	drastically different constructs. Their analytic approach seems to 'jam' these disparate ideas together without a justification. MAJOR COMMENT: I am particularly concerned about including medical uncertainty as a "sensitive" topic. Uncertainty is ubiquitous in medical practice, and conceptually, I don't believe it needs to be treated as a sensitive subject. Indeed, there is some evidence that patients find value and comfort when their doctors openly discuss uncertainty with them. (PMID 35366717) Major Comment: I have serious concerns about the validity and reliability of their data analysis. Many of their outcomes did not include objective definitions and would appear highly prone to subjective interpretation. For example, how did the researchers categorize sensitive topics? Were they provided definitions? Was there rater training? Any means to ensure reliability? Likewise, many of their outcome measures seem highly subjective (e.g., "The study team rated objective knowledge by comparing patients' recall of information with... medical information from the chart). Minor Comment: The authors highlight the "strong association" between sensitive topics and patient satisfaction throughout their manuscript. However, based on my calculation of their results, the Cohen's d effect size is only 0.18, indicating a small difference in satisfaction between sensitive and no sensitive topics. Is this difference clinically relevant? Minor Comment: The authors include many results in their discussion section (e.g., Box 1 and 2). They should be placed in the results section. Also, they don't provide any details of their "qualitative analysis".
--	--

REVIEWER	palese, alvisa Univ Udine
REVIEW RETURNED	25-Jun-2023

GENERAL COMMENTS	I congratulate the theme which is really very interesting. I have the following suggestions (a) in the abstract, where you indicate that it is a secondary analysis, also indicate the registration of the first trial. (b) between the strengths and weaknesses, the first sentence needs to be changed: it is not the first trial on sensitive topics, but it is the first secondary analysis on this topic. (c) in the literature review we suggest that you consider patient perception; recent metasynthesis (Bressan V, Cadorin L, Stevanin S, Palese A. Patients experiences of bedside handover: findings from a meta-synthesis. Scand J Caring Sci. 2019 Sep;33(3):556-568. doi: 10.1111/scs .12673. Epub 2019 Mar 13. PMID: 30866081) have summarized the studies in this direction. It might be useful to place the study in a broader context of the debate on this issue. (d) the concept of ward round is not very clear - to ensure a visible and transparent evaluation of what has been done (each country has its own traditions regarding patient rounds), I suggest you consider a more precise operational description and conceptual of this concept. An overview of the possible operational and conceptual bases has been recently illustrated (Bayram A et al . Untangling a complex skein on ward round terminologies, purposes, and main features through a rapid review. J Adv Nurs. 2023 Jun 14. doi: 10.1111 /jan.15732. Epub ahead of print. PMID:
--

	37313993). In fact, the paper mentions 'standard practice - page 8' but it is not clear what is meant by standard practices. It should also be clarified who the participants were (you are referring to a group of professionals') and how many there were. (e) the categorization of sensitive topics is not clear and needs to be developed better. This is a very interesting aspect. (f) the results are interesting. (g) in discussions, mention 'ancillary project' while presenting the study as a secondary analysis. In the discussion I would suggest not to indicate examples (of data collection, see Box 1, 2) and to focus your attention on the results already presented. You can take these examples to the supplementary tables; the study is already very rich and I would not force readers to reconsider the results as they appreciate your discussion. (h) within limits, I suggest that you consider the use of the VAS differently: you used a one-dimensional metric system to evaluate satisfaction with care that reflects a very general concept (you did not measure satisfaction with medical care or the ward round but towards 'the cures'). This is an important limitation that needs to be highlighted. (i) the conclusions perhaps need to be strengthened. It is not only a question of improving communication skills but also of choosing whether or not these sensitive topics should be discussed at the bedside. The setting is also important and perhaps doctors should be helped to understand discussing them in other settings and contexts. I really congratulate you and I hope that these stimuli can be of help.
--	--

REVIEWER	Valente, Roberto University College Hospital
REVIEW RETURNED	27-Jun-2023

GENERAL COMMENTS	The authors report a quali-quantitative analysis of the impact on patient care satisfaction of discussing sensitive topics during internal medicine ward rounds, by analysing audiotapes recorded during a previous study. The topic appears to be a crucial domain in communication to patients, with significant impact on their care satisfaction. Overall this seems to me a well designed, delivered and reported work. However, I would suggest revising further some parts of the manuscripts prior to considering its publication. ABSTRACT I would suggest minor re-writing by an expert medical scientific writing, in order to make it more understandable and easy to read.  1. The objectives are not easy to understand: what is the objective of the previous study, and what the objective of the current one? How do the two studies differ one from the other? 2. There are some typo errors in the draft, such as the use of brackets. 3. Conclusions appear too peremptory and universal. A start of the first sentence such as "In our patient population" etc. might be
---

	more appropriate, given the limitations mentioned in the manuscript in addition to those carried by in the study design. INTRODUCTION Please make sure the difference of this study from the previous linked ones is clearly explained, and what the current study adds to such previous ones, as it is not entirely clear to me at this point. This needs explicit clarification. In the last 'aims' paragraph the 4 reference is presented. What is the reason of showing this reference? METHODS The labelling description of the present study as an "ancillary analysis" (i.e. in the outcome measures section) is misleading to me. Is it not the the manuscript object? RESULTS Flawless description. Brackets in the second "baseline ..." paragraph are inconsistent, please amend. DISCUSSION I would suggest mentioning here in a short paragraph why the VAS and patient satisfaction (and patient information self assessment) measure rather than Likert scaling Page 24, lines 21-26 paragraph is not easy to understand to me. Please could you re-write? Page 25, typo on line 38: "heree" CONCLUSION I would remove "ancillary" here. This study might be ancillary to the main RCT, however is the object of the current manuscript.
--	---

VERSION 1 – AUTHOR RESPONSE

Reviewer: 1

Dr. Thurayya Eid, King Saud University

Comments to the Author:

Minor revision

page 6 line 59 who approached the patients during recruitment

Reply: We now clarified this point as follows: "Newly admitted adult inpatients on medical wards expecting their first weekly ward round consultation were approached by a member of the study team regarding inclusion."

page 7 line 54 the "round" not "Rond"

Reply: Thank you we now corrected this typo.

page 7 line 53 name the manufactures of. the voice recording software

Reply: We now added the voice recording software, which was the Apple iPad device-internal

software 'Voice Memo', to the method section.

page 7 line 45 at what time the ward rounds usually occurs. in Australia for example the round time occurs from 09:00 - 11:00 am

Reply: Thank you for your question. We now added the ward round time frame to the data collection section as requested. It now reads: "During the ward rounds, which were conducted between 9 and 11 a.m., an observer from the research team was present to document timing (i.e., the duration of the ward round allocated per patient)."

page 9 line 3 the word "instances"

Page 24 line 51" ward round"

page 24 line 53 "differential"

Reply: Thank you- we now corrected these typos.

Reviewer: 2

Dr. John Ratelle, Mayo Clinic

Comments to the Author:

Thank you for the opportunity to review this manuscript. I commend the authors for exploring the important and patient-centered topic of sensitive issues raised during hospital ward rounds. However, I believe there are serious conceptual and methodological issues that threaten the validity of their findings.

Major comment: The authors' literature review is insufficient. At several points in the manuscript, they highlight the novelty of their study, the first to "analyze communication regarding sensitive topics during ward rounds.". This claim is categorically false. For example, Jason Satterfield and colleagues studied sensitive topics during ward rounds nearly a decade ago (ref: Acad Med. 2014 Nov;89(11):1548-57. PMID 25250757)

Reply: We agree and now included the proposed reference in the introduction section and remove the above mentioned statement.

Major comment: The authors have not provided an adequate conceptual or operational definition of "sensitive" topics. For example, medical uncertainty and patient non-adherence are drastically different constructs. Their analytic approach seems to 'jam' these disparate ideas together without a justification.

Reply: We agree and now provide a better definition and justification for "delicate topics" as follows: "Because, to our knowledge, there is no well accepted definition of sensitive topics, we defined "sensitive topics" based on the clinical experience of the physician–researcher team and by reviewing previous literature. Sensitive topics were coded prospectively as a situation where at least one of the follow topics was discussed with the patient during the ward round: medical uncertainty, psychiatric comorbidities, tumor diagnosis, social issues, non-adherence, previous conflicts between patient and treating team, and treatment failure."

We also added the following bullet point to the limitation section: "Different sensitive topics have been pooled into a combined variable for the purpose of this analysis."

MAJOR COMMENT: I am particularly concerned about including medical uncertainty as a "sensitive" topic. Uncertainty is ubiquitous in medical practice, and conceptually, I don't believe it needs to be

treated as a sensitive subject. Indeed, there is some evidence that patients find value and comfort when their doctors openly discuss uncertainty with them. (PMID 35366717)

Reply: Thank you for pointing out this important issue. We now performed a sensitivity analysis excluding uncertainty from the definition of “sensitive topics”. Interestingly, the models showed very robust and similar results regarding the different associations. We now added this information to the manuscript as follows: "Finally, there is no well accepted definition of sensitive topics and we thus defined it based on the clinical experience of the physician-researcher team. Still, there is some data suggesting that patients find value and comfort when their doctors openly discuss uncertainty with them. (REF PMID 35366717). Yet, a sensitivity analysis excluding uncertainty from the definition of sensitive topics showed similar results. "

Major Comment: I have serious concerns about the validity and reliability of their data analysis. Many of their outcomes did not include objective definitions and would appear highly prone to subjective interpretation. For example, how did the researchers categorize sensitive topics? Were they provided definitions? Was there rater training? Any means to ensure reliability? Likewise, many of their outcome measures seem highly subjective (e.g., “The study team rated objective knowledge by comparing patients’ recall of information with... medical information from the chart).

Reply: Thank you. We agree that outcomes and variables in the trial were often subjective, which is a concern for most studies focusing on PROMS. We did in fact train all raters and gave them guidance on the definition of the different variables and also held regular training and discussion sessions. We now provide a more clear definition of sensitive topics and also included a limitation in this regard as outlined above.

Minor Comment: The authors highlight the “strong association” between sensitive topics and patient satisfaction throughout their manuscript. However, based on my calculation of their results, the Cohen’s d effect size is only 0.18, indicating a small difference in satisfaction between sensitive and no sensitive topics. Is this difference clinically relevant?

Reply: Thank you for your comment. We agree with your concern and now remove the word “strong” as indeed the difference in satisfaction seems to be rather small. However, satisfaction-related measures usually are skewed and show the present ceiling-effect. Therefore, we consider the difference to be clinically relevant. However, we agree that satisfaction measures are a rather vague concept and may thus not be an ideal variable.

Minor Comment: The authors include many results in their discussion section (e.g., Box 1 and 2). They should be placed in the results section. Also, they don’t provide any details of their “qualitative analysis”.

Reply: Thank you very much for this comment. The boxes represent samples of the qualitative analysis. The entire analysis can be found in the supplement.

Reviewer: 3

Dr. alvisa palese, Univ Udine

Comments to the Author:

I congratulate the theme which is really very interesting. I have the following suggestions

(a) in the abstract, where you indicate that it is a secondary analysis, also indicate the registration of the first trial.

Reply: Thank you for your suggestion. We added the Trial registration in the abstract.

(b) between the strengths and weaknesses, the first sentence needs to be changed: it is not the first trial on sensitive topics, but it is the first secondary analysis on this topic.

Reply: Thank you - we agree and replaced the first bullet point (see Reviewer 2).

(c) in the literature review we suggest that you consider patient perception; recent metasynthesis (Bressan V, Cadorin L, Stevanin S, Palese A. Patients experiences of bedside handover: findings from a meta-synthesis. *Scand J Caring Sci.* 2019 Sep;33(3):556-568. doi: 10.1111/scs .12673. Epub 2019 Mar 13. PMID: 30866081) have summarized the studies in this direction. It might be useful to place the study in a broader context of the debate on this issue.

Reply: Thank you very much for proposing this interesting meta-analysis. We placed the relevant findings of the study in the introduction section as follows “There is important work about the experiences of patients regarding bedside shift suggesting that asking patient consent, discussing potential critical issues and the degree of involvement preferred at hospital admission is strongly recommended.”

(d) the concept of ward round is not very clear - to ensure a visible and transparent evaluation of what has been done (each country has its own traditions regarding patient rounds), I suggest you consider a more precise operational description and conceptual of this concept. An overview of the possible operational and conceptual bases has been recently illustrated (Bayram A et al . Untangling a complex skein on ward round terminologies, purposes, and main features through a rapid review. *J Adv Nurs.* 2023 Jun 14. doi: 10.1111 /jan.15732. Epub ahead of print. PMID: 37313993). In fact, the paper mentions 'starad practice - page 8' but it is not clear what is meant by standard practices. It should also be clarified who the participants were (you are referring to a group of professionals') and how many there were.

Reply: Thank you. Indeed we provided a detailed description of the ward round structure in the original Annals paper and now also gave more details in the current manuscript referring to the original trial as follows: “Details of the study intervention and a detailed description of the ward round procedure have been reported earlier.¹ In brief, for the purpose of standardization and in line with current practice in Switzerland, the ward round followed the routine medical ward round procedures in both groups, with defined roles of physicians and nurses per usual practice in each participating hospital. In the bedside presentation group, case presentations and discussions occurred only at the bedside in front of the patient, including clinical examination as appropriate, with no discussions beforehand. In the outside the room group, case presentation and discussions were primarily held in the hallway outside the room without the patient present. Afterwards, the team entered the room and gave the patient a short summary of the medical situation, completed the gathering of medical information, examined the patient as needed, and discussed the next steps. Patients, study coordinators, and treating clinicians were not blinded to the allocation. However, study investigators involved in a patient’s outcome assessment were blinded to trial allocation.”

(e) the categorization of sensitive topics is not clear and needs to be developed better. This is a very interesting aspect.

Reply: Thank you for this input. We agree and as described above (see Reviewer 2), we now provide further information about the definition process of sensitive topics to the method section and added the pooling of sensitive topics to the limitation section.

(f) the results are interesting.
Reply: Thank you very much.

(g) in discussions, mention 'ancillary project' while presenting the study as a secondary analysis. In the discussion I would suggest not to indicate examples (of data collection, see Box 1, 2) and to focus your attention on the results already presented. You can take these examples to the supplementary tables; the study is already very rich and I would not force readers to reconsider the results as they appreciate your discussion.

Reply: Thank you. Because the definition of sensitive topics is rather new to many readers, we also wanted to give some qualitative aspect to the manuscript. However, most of this analysis is only presented in the appendix.

(h) within limits, I suggest that you consider the use of the VAS differently: you used a one-dimensional metric system to evaluate satisfaction with care that reflects a very general concept (you did not measure satisfaction with medical care or the ward round but towards 'the cures'). This is an important limitation that needs to be highlighted.

Reply: Thank you very much for your input. We agree that satisfaction measures are a rather vague concept and may thus not be an ideal variable. However, in order to cover as many aspects of satisfaction as possible, we defined the primary endpoint "patient's overall satisfaction" as the mean of different satisfaction aspects, i.e. satisfaction with ward rounds, hospital stay, medical care, physician communication, and nursing team communication, all measured on a VAS from 0-100. This is the definition that was predefined and also used for the Annals paper. We thus aimed to stay consistent with the initial analysis.

(i) the conclusions perhaps need to be strengthened. It is not only a question of improving communication skills but also of choosing whether or not these sensitive topics should be discussed at the bedside. The setting is also important and perhaps doctors should be helped to understand discussing them in other settings and contexts.

Reply: Thank you for this suggestion. We agree and added your point to the conclusion.

I really congratulate you and I hope that these stimuli can be of help.

Reply: Thank you.

Reviewer: 4

Mr. Roberto Valente, University College Hospital, IRCCS Ospedale Policlinico San Martino

Comments to the Author:

The authors report a quali-quantitative analysis of the impact on patient care satisfaction of discussing sensitive topics during internal medicine ward rounds, by analysing audiotapes recorded during a previous study.

The topic appears to be a crucial domain in communication to patients, with significant impact on their care satisfaction.

Overall this seems to me a well designed, delivered and reported work. However, I would suggest revising further some parts of the manuscripts prior to considering its publication.

ABSTRACT

I would suggest minor re-writing by an expert medical scientific writing, in order to make it more

understandable and easy to read.

1. The objectives are not easy to understand: what is the objective of the previous study, and what the objective of the current one? How do the two studies differ one from the other?

Reply: Thank you. We agree that the objective of the original trial may not have been entirely comprehensible. Thus, as described above (see editors' comments) we changed the introduction for a better understanding of the main study. We also did some rewriting to be clearer in our message.

2. There are some typo errors in the draft, such as the use of brackets.

Reply: Thank you very much for reading the manuscript carefully. We corrected the issues with brackets.

3. Conclusions appear too peremptory and universal. A start of the first sentence such as "In our patient population" etc. might be more appropriate, given the limitations mentioned in the manuscript in addition to those carried by in the study design.

Reply: Thank you. We agree and revised the first sentence of the conclusion. Further, we have expanded the conclusion to make it less peremptory and to emphasize the need for further research.

INTRODUCTION

Please make sure the difference of this study from the previous linked ones is clearly explained, and what the current study adds to such previous ones, as it is not entirely clear to me at this point. This needs explicit clarification.

Reply: Thank you. We now provide a description of the original trial's concept and results in the introduction section, which shall clarify what this ancillary analysis adds to the previous study.

In the last 'aims' paragraph the 4 reference is presented. What is the reason of showing this reference?

Reply: Thank you for your question. We removed the reference.

METHODS

The labelling description of the present study as an "ancillary analysis" (i.e. in the outcome measures section) is misleading to me. Is it not the manuscript object?

Reply: Thank you we now changed the wording as suggested by the reviewer and the editor.

RESULTS

Flawless description. Brackets in the second "baseline ..." paragraph are inconsistent, please amend.

Reply: Thank you. We unified the brackets.

DISCUSSION

I would suggest mentioning here in a short paragraph why the VAS and patient satisfaction (and patient information self assessment) measure rather than Likert scaling

Reply: The definition of satisfaction was predefined and also used for the Annals paper. We thus

aimed to stay consistent with the initial analysis.

Page 24, lines 21-26 paragraph is not easy to understand to me. Please could you re-write?

Reply: Thank you for this comment. We adjusted this paragraph.

Page 25, typo on line 38: "heree"

Reply: Thank you for the correction. We corrected the mistake.

CONCLUSION

I would remove "ancillary" here. This study might be ancillary to the main RCT, however is the object of the current manuscript.

Reply: Thank you for your suggestion. We agree and removed the word.

VERSION 2 – REVIEW

REVIEWER	Ratelle, John Mayo Clinic
REVIEW RETURNED	14-Aug-2023

GENERAL COMMENTS	Thank you to the authors for addressing my concerns. I am overall satisfied with their revisions. My final request is to update the "Strengths and Limitations" section under the abstract. Specifically, it would be beneficial to clarify that the lack of a universally agreed-upon definition for "sensitive topics" is a limitation of this study. This ambiguity can lead to different researchers defining and categorizing these topics in various ways, which may result in inconsistent results and conclusions. The current statement, "Different sensitive topics have been pooled into a combined variable for the purpose of this analysis," is unclear in its portrayal of whether this is a strength or limitation of the study.
--

REVIEWER	palese, alvisa Univ Udine
REVIEW RETURNED	28-Jul-2023

GENERAL COMMENTS	Congratulations
-----------------